# RETRACTED: Ascertaining the Influence of Lacto-Fermentation on Changes in Bovine Colostrum Amino and Fatty Acid Profiles

**DOI:** 10.3390/ani13193154

**Published:** 2023-10-09

**Authors:** Vytautė Starkutė, Ernestas Mockus, Dovilė Klupšaitė, Eglė Zokaitytė, Saulius Tušas, Ramutė Mišeikienė, Rolandas Stankevičius, João Miguel Rocha, Elena Bartkienė

**Affiliations:** 1Institute of Animal Rearing Technologies, Lithuanian University of Health Sciences, Tilzes St. 18, LT-47181 Kaunas, Lithuania; vytaute.starkute@lsmu.lt (V.S.); saulius.tusas@lsmu.lt (S.T.); ramute.miseikiene@lsmu.lt (R.M.); 2Department of Food Safety and Quality, Lithuanian University of Health Sciences, Tilzes St. 18, LT-47181 Kaunas, Lithuania; 3Department of Animal Nutrition, Lithuanian University of Health Sciences, Tilzes St. 18, LT-47181 Kaunas, Lithuania; 4Universidade Católica Portuguesa, CBQF—Centro de Biotecnologia e Química Fina—Laboratório Associado, Escola Superior de Biotecnologia, Rua Diogo Botelho 1327, 4169-005 Porto, Portugal; 5Laboratory for Process Engineering, Environment, Biotechnology and Energy (LEPABE), Faculty of Engineering, University of Porto (FEUP), Rua Dr. Roberto Frias, s/n, 4200-465 Porto, Portugal; 6Associate Laboratory in Chemical Engineering (ALiCE), Faculty of Engineering, University of Porto (FEUP), Rua Dr. Roberto Frias, s/n, 4200-465 Porto, Portugal

**Keywords:** bovine colostrum, fermentation, lactic acid bacteria, amino acids, fatty acids, proteomic profiles, lipidomic profiles, biogenic amines

## Abstract

**Simple Summary:**

Bovine colostrum (BCOL) is rich in functional molecular ingredients, which can be used in human nutrition. However, the lack of data concerning changes in the main nutrients of colostrum during technological processing often limits its application at an industrial scale. Moreover, using colostrum for food enrichment or supplement production requires considering the changes in colostrum constituents that may occur during processing. The aim of this study was to collect samples of BCOL from different agricultural companies (A, B, C, D and E) in Lithuania and to evaluate the influence of lacto-fermentation with *Lactiplantibacillus plantarum* strain 135 and *Lacticaseibacillus paracasei* strain 244 on bovine colostrum amino (AA), biogenic amine (BA) and fatty acid (FA) profiles. It was established that the composition of AA and FA profiles in bovine colostrum varied, and these differences are related to both the source of colostrum samples and lactic acid bacteria (LAB) used for bovine colostrum fermentation. Moreover, LAB fermentation leads to different BA formations during the process. Finally, the utilization of bovine colostrum proved to be challenging because of the variability in its composition. For successful bovine colostrum application at an industrial scale, its composition should be adjusted starting from the primary production.

**Abstract:**

The aim of this study was to collect samples of bovine colostrum (BCOL) from different sources (agricultural companies A, B, C, D and E) in Lithuania and to ascertain the influence of lacto-fermentation with *Lactiplantibacillus plantarum* strain 135 and *Lacticaseibacillus paracasei* strain 244 on the changes in bovine colostrum amino (AA), biogenic amine (BA), and fatty acid (FA) profiles. It was established that the source of the bovine colostrum, the used LAB, and their interaction had significant effects (*p* < 0.05) on AA contents; lactic acid bacteria (LAB) used for fermentation was a significant factor for aspartic acid, threonine, glycine, alanine, methionine, phenylalanine, lysine, histidine, and tyrosine; and these factor’s interaction is significant on most of the detected AA concentrations. Total BA content showed significant correlations with glutamic acid, serine, aspartic acid, valine, methionine, phenylalanine, histidine, and gamma amino-butyric acid content in bovine colostrum. Despite the differences in individual FA contents in bovine colostrum, significant differences were not found in total saturated (SFA), monounsaturated (MUFA), and polyunsaturated (PUFA) fatty acids. Finally, the utilization of bovine colostrum proved to be challenging because of the variability on its composition. These results suggest that processing bovine colostrum into value-added formulations for human consumption requires the adjustment of its composition since the primary production stage. Consequently, animal rearing should be considered in the employed bovine colostrum processing technologies.

## 1. Introduction

Bovine colostrum is commonly known as the life’s first food. It is also called “liquid gold” or “immune milk” [1,2,3], and it is recognized as being rich in functional molecules [4,5,6,7]. Additionally, colostrum contains valuable nutrients (amino and fatty acids, as well as carbohydrates) [3]. However, their contents can be affected by many factors [8,9,10,11,12,13,14,15], including the technological processing. Therefore, using colostrum for food enrichment or supplement production requires consideration of the variability in colostrum constituents that may occur during processing. Our previous studies showed that fermentation with selected lactic acid bacteria (LAB) strains can be applied to reduce undesired microorganisms present in bovine colostrum (microbial decontamination) [14], as well as to drive out to a broader spectrum of antimicrobial properties in this valuable and prospective food or supplement ingredient [15]. Indeed, technologically functionalized colostrum can be successfully applied to the development of attractive, functional product formulations [16,17,18].

Pre-treatment (including lacto-fermentation) can lead to changes in the proteomic and lipidomic profiles of the colostrum, and these changes are not always desirable. For instance, they can lead to the formation of undesired biogenic amines (BA). Although the formation of BA is crucial for the proper course of the metabolic processes of the organisms, its excessive concentration has been proved to cause diarrhea, food poisoning, vomiting, sweating, or tachycardia and can accelerate carcinogenesis [19].

Although the consumption of colostrum is generally considered safe for human nutrition [3], changes during the technological processes must be considered. During the collection of bovine colostrum from agricultural companies and its processing, it is very important to ascertain the changes in its main composition. The changes in bovine colostrum composition can vary not only because of the processing technology under use (e.g., the type of microorganisms used for fermentation) but also because of the animal-rearing technologies employed in the farms, the hygienic conditions, and good farming practices. All these factors are related to the primary constituents of colostrum, including the chemical and microbiological profiles.

It was reported that the total protein content in bovine colostrum (on the first day) is, on average, 15% [20]. Although there are detailed studies on the composition of proteins in bovine milk at different lactation stages [21,22,23,24,25], there are relatively few studies concerning the changes in the amino acid profiles [26]. Previous studies on milk metabolomics have been focused on the comparison of free amino acids and insoluble-proteome amino acids in bovine colostrum and mature milk, as well as their metabolic pathways during lactation [26]. In this present study, the main idea was to collect representative samples of bovine colostrum from different agricultural companies in different geographical regions of Lithuania, followed by fermentation of the collected samples using the microbial starter cultures *Lactiplantibacillus plantarum* 135 and *Lacticaseibacillus paracasei* 244, and then examine the changes in bovine colostrum amino acid and fatty acid profiles. The main hypothesis of this study was that the changes in bovine colostrum amino and fatty acid profiles during fermentation could be related not only to the starter culture strain under use but also to the specific composition of bovine colostrum, which varied in different agricultural companies. Additionally, as an indicator of amino acid degradation, biogenic amine content in bovine colostrum was studied, and the relationships between amino acid content and BA were studied.

The fat composition of colostrum during the fermentation also varies [14]. Considering that colostrum contains a higher percentage of fat than regular milk [27], the evaluation of the changes in fatty acid profile during lactic acid fermentation becomes very important. It was reported that the colostrum fat profile contains higher concentrations of palmitic, palmitoleic, and myristic acids than regular milk [28].

Finally, this study aimed to evaluate the influence of lacto-fermentation on the changes in amino and fatty acid profiles of bovine colostrum. Additionally, the BA profile was evaluated, and its correlation with amino acid contents was studied. The main parameters indicating the fermentation efficiency were analyzed to ascertain the effectiveness of the LAB fermentation process, and the changes in pH values and LAB viable counts in bovine colostrum samples were evaluated. The obtained results provide more accurate data regarding bovine colostrum amino and fatty acid profiles. This information can be applied from two different perspectives, viz., the colostrum preservation in farms as well as the preparation of nutraceuticals for human nutrition. Such a valuable data base can be used for planning further applications of colostrum as a valuable food ingredient at an industrial scale.

## 2. Materials and Methods

### 2.1. Materials

The bovine colostrum (BCOL) samples were collected from five agricultural companies (farms) in Lithuania keeping Holstein dairy cows during the spring period of the year 2022: (A) Dauksiai, (B) Sauselio village, (C) Geluvos village, (D) Paliepiu village, and (E) Alksnupiu village. Samples (5% of the cow population from each company) were collected from the multiparous cows of the Holstein breed, which is the most common dairy breed in the country. BCOL samples were collected in the agricultural companies within 2 h of calf delivery, placed in 1–5 L volume plastic bags, and stored at −18 °C in a freezer before use. It should be mentioned that this is a usual practice at dairy cow farms.

Two LAB strains (liquid form) [Lp. plantarum (135/LUHS135) and *Lc. paracasei* (244/LUHS244] were obtained from the Lithuanian University of Health Sciences collection (Kaunas, Lithuania) and selected for *BCOL* fermentation, taking into account their ability to ferment lactose [29]. Before the experiment, the selected LAB strains were separately stored at −80 °C in a Microbank system (Pro-Lab Diagnostics, Bromborough, UK) and grown on de Man, Rogosa, Sharpe (MRS) broth (CM 0359, Oxoid, Hampshire, UK) at 30 °C for 48 h prior to use [30].

### 2.2. Fermentation of Bovine Colostrum (BCOL)

Three milliliters of MRS broth, in which the two LAB strains were individually multiplied [average LAB cell concentration of 9.2 log_10_ colony-forming units (CFU)/mL] were used to inoculate 100 mL of defrosted [at (24 ± 2) °C for 12 h] BCOL, followed by fermentation for 24 h at (30 ± 2) °C.

In total, 15 BCOL treatments were obtained: non-fermented BCOL—(A), (B), (C), (D) and (E); BCOL treated (fermented) with LAB 135 strain (*Lp. plantarum* 135)—A135, B135, C135, D135, E135; and BCOL treated (fermented) with LAB 244 strain (*Lc. paracasei* 244)—A244, B244, C244, D244, E244. The experimental design is depicted in Figure 1.

The number of BCOL samples collected from each source of colostrum was determined based on the total number of keeping cows. In summary, the number of samples collected form agriculture companies with 500 (A), 120 (B), 159 (C), 131 (D), and 1403 (E) cows were 25, 6, 7, 5, and 70, respectively. The farms were selected based on similar feeding and housing conditions. Animals were kept loose on rubber mats all year round and milked twice a day in a milking parlor. The cows received a total mixed diet based on grass and maize silage, hay, and concentrate. There were no further specifications regarding udder health parameters, such as the incidence of clinical mastitis.

### 2.3. Determination of pH and Lactic Acid Bacteria (LAB) Viable Counts in Bovine Colostrum (BCOL) Samples

The pH of BCOL was evaluated with a pH meter (Inolab 3, Hanna Instruments, Venet, Italy) by inserting the pH electrode into the samples.

For the evaluation of LAB viable counts, 10 g of BCOL was homogenized with 90 mL of 9 g/L NaCl aqueous solution. Serial decimal dilutions from 10^−4^ to 10^−8^ were used for sample preparation. Sterile MRS agar (CM0361, Oxoid) of 5 mm thickness on Petri dishes was used for bacterial inoculation. Petri dishes were separately inoculated with the BCOL suspension using the surface spread-plate technique and were incubated under anaerobic conditions at (30 ± 2) °C for 72 h. The number of viable LAB counts was determined in the dilutions containing between 30 and 300 colonies and expressed as log_10_ of colony-forming units per mL (CFU/mL).

### 2.4. Analysis of Amino Acid (AA) Profile in Bovine Colostrum (BCOL) Samples

Sample preparation and dansylation were performed according to the method of Ben-Gigirey et al. (1999) [31], with some modifications [30]. A homogenous sample (target of ~1 g) was weighed in a 15 mL sample tube, and the analytes were extracted with 10 mL of aqueous 0.1 M HCl solution by shaking for 1 h. The resultant mixture was centrifuged at 4000 RPM for 5 min. For derivatization, 200 µL of resultant supernatant was diluted to 500 µL with 0.1 M HCl solution. The subsequent mixture was made alkaline by adding 40 µL of 2M NaOH and 70 µL of saturated NaHCO_3_ solution. Derivatization was performed by adding 0.5 mL of 20 mg/mL dansyl chloride solution in acetonitrile and heating the resulting mixture at 60 °C for 30 min. The reaction mixture was cooled down to room temperature, centrifuged at 12,000 rpm for 5 min, and filtered through a 0.22 µm membrane filter to the autosampler vial. The concentration of analytes was determined using the Varian ProStar HPLC system (Varian Corp., Palo Alto, CA, USA): two ProStar 210 pumps, a ProStar 410 autosampler, and a Thermo Scientific LCQ Fleet Ion trap mass detector. For analyte detection, a mass spectrometer operated in positive ionization consecutive reaction monitoring mode for specific ions (which correspond to the derivatized analyte). The concentration of the analyte was determined from a calibration curve, which was obtained by derivatizing analytes at different concentrations. For the separation of derivatives, a Discovery^®^ HS C18 column (150 mm length × 4.6 mm diameter, 5 µm diameter particle size; SupelcoTM Analytical, Bellefonte, PA, USA) was used. The mobile phase A was 0.1% formic acid in 5% aqueous acetonitrile, and the phase B was 0.1% in acetonitrile. A 0.3 mL/min flow-rate was used for the analysis, and a 10 µL injection volume was applied. Analytical gradient was as follows: from 0 to 1 min (linear gradient) with 15 to 50% B (flow-rate of 1 mL/min), from 3 to 3.5 min (flow-rate of 1.0 to 0.3 mL/min), from 3.5 to 10 min (linear gradient) with 50 to 70% B, from 10 to 15 min (linear gradient) with 70 to 90% B, from 15 to 30 min with 90 to 95% B, followed by re-equilibration for 11 min with 15% B (flow-rate increased to 0.8 mL/min).

### 2.5. Analysis of Biogenic Amine (BA) Profile in Bovine Colostrum (BCOL) Samples

The extraction and determination of BA in non-fermented and fermented BCOL followed the procedures developed by Ben-Gigirey et al. [32] with some modifications as described by Bartkiene et al. [15]. Derivatization of BCOL was performed with dansyl chloride. A Varian ProStar HPLC system (Varian Corp., Palo Alto, CA, USA) equipped with a ProStar 325 ultra-violet/visible (UV/VIS) detector and Galaxy software (Agilent, Santa Clara, CA, USA; software version 1.9.3.2, Varian Inc., Palo Alto, CA, USA) was employed. A Discovery^®^ HS C18 column 150 mm length × 4.6 mm diameter, 5 μm diameter particle size (SupelcoTM Analytical, Bellefonte, PA, USA) was used. The detection limit of BA was 0.1 mg/kg.

### 2.6. Analysis of Fatty Acid (FA) Profile in Bovine Colostrum (BCOL) Samples

The extraction of lipids for fatty acid (FA) quantification was undertaken with chloroform/methanol (2:1, *v*/*v*), and fatty acid methyl esters (FAME) were prepared according to the protocol described by Pérez-Palacios et al. [33]. The FA composition in BCOL was identified using a gas chromatograph GC-2010 Plus (Shimadzu Europa GmbH, Duisburg, Germany) equipped with a mass spectrometer GC/MS-QP2010 (Shimadzu Europa GmbH, Duisburg, Germany). Separation was carried out on a Stabilwax-MS column (30 m length × 0.25 mm diameter and 0.25 μm diameter particle size) (Restek Corporation, Bellefonte, PA, USA). The mass spectrometer operated at full scan mode, and the analyte was injected in split mode at a 1:60 split ratio. The following parameters were used: MS ion source temperature: 240 °C; MS interface temperature 240 °C; helium (carrier gas) flow-rate: 0.90 mL/min; injector temperature: 240 °C; and oven temperature program: 50 °C (4 min), 10 °C/min to 110 °C (1 min), 15 °C/min to 160 °C (2 min), 2.5 °C/min to 195 °C (1 min), 2 °C/min to 230 °C (1 min), and 2 °C/min to 240 °C (12 min). The individual FAME peaks were identified by comparing their retention times with FAME standards (Merck & Co., Inc., Kenilworth, NJ, USA). The quantification was determined using the corrected area normalization. 

### 2.7. Statistical Analysis

The results were expressed as independent mean value ± standard error (SE) of the BCOL collected from each source. As previously stated, BCOL was collected from, on average, 5% of the dairy cows to obtain samplings. All BCOL from each source of colostrum were analyzed separately. The analyses of independent BCOL were repeated in triplicate, and three repetitions were averaged before analysis. Data were analyzed using multivariate analysis of variance to evaluate the effects of different agricultural companies, different LAB strains used for fermentation, and the interaction between these factors on the BCOL parameters (V22.0, IBM SPSS Statistics, Chicago, IL, USA, 2013). Moreover, Pearson correlations were calculated between the parameters of BCOL, and the strength of the correlation was interpreted according to Evans [34]. The results were recognized as statistically significant at *p* ≤ 0.05.

## 3. Results and Discussion

### 3.1. Influence of Fermentation on pH and Lactic Acid Bacteria (LAB) Viable Counts in Bovine Colostrum (BCOL)

The main parameters of the fermentation performance (pH values and LAB viable counts) are shown in Table 1. No significant differences (*p* > 0.05) were observed in the values of the pH and LAB viable counts in the same BCOL group (non-fermented, fermented with LAB strain 135, or fermented with the LAB strain 244). In addition, no statistically significant differences (*p* > 0.05) were also observed in the LAB viable counts of BCOL between groups of non-fermented, fermented with LAB strain 135, and fermented with LAB strain 244 samples. However, as expected, the BCOL samples fermented with LAB strains 135 and 244 showed significantly lower (*p* ≤ 0.05) pH values (on average, by 30.2 and 30.6%, respectively) in comparison with non-fermented ones. A significant correlation between pH and LAB viable counts of BCOL was not established. However, tests of between-subject effects showed that the analyzed factors (different agricultural companies and different LAB strains used for fermentation), as well as the interaction between these factors, were statistically significant on pH values of BCOL (*p* = 0.010, *p* < 0.001 and *p* = 0.030, respectively).

It was reported that the pH value of BCOL was lower than that of milk [35]. Moreover, at parturition, the pH could vary between 6.0 and 6.6 [36], and it increased with post-partum time [7,37,38,39]. It is expected that the pH values of colostrum are lower in comparison with milk because BCOL contains blood components, and its pH is closer to that of blood (on average, 7.40) [40]. In addition, BCOL contains dihydrogen phosphate, citrate, and carbon dioxide, and these compounds are also associated with the lower pH found in colostrum [41]. Additionally, BCOL contains high numbers of LAB, which usually possess probiotic functions [42]. The nutrients present in the composition of BCOL can promote the proliferation of naturally occurring LAB in BCOL and may lead to a drop in colostrum pH as well as to the reduction of the energy value [43]. The observed changes in the pH and acidity depend on the substrate composition [44,45], and this specificity can be associated with the content of fermentable sugars in the BCOL. The main carbohydrates in BCOL are lactose, oligosaccharides, glycolipids, glycoproteins, and nucleotide sugars [20]. Lactose content in BCOL is, on average, 2.5% [46]. However, lactose content can vary in relation to numerous different factors. It was reported that the lactose concentration in BCOL can be, on average, 1.2% [46,47]. In addition to lactose, BCOL contains glucose, fructose, glucosamine, galactosamine, N-acetylneuraminic acid, and oligosaccharides [20]. The concentration of the latter sugars in BCOL can also vary [48,49,50,51,52,53]. A priori, no differences in the LAB viable counts were expected. Finally, in the current study, we observed non-significant differences (*p* > 0.05) in the pH and LAB viable counts of BCOL when using two different starter cultures for BCOL fermentation (regarding the pH of fermented samples and LAB viable counts in all groups). Nevertheless, fermentation with LAB strain 135 and 244 significantly reduced the pH of the samples. Further studies with the aim to analyze the metagenomic profiles of fresh colostrum as well as their changes during fermentation process are likely to become very prospective because of the actual absence of information concerning possible changes to the microbial community in BCOL during fermentations with selected LAB strains.

### 3.2. Amino Acid (AA) and Gamma-Aminobutyric Acid (GABA) Content in Non-Fermented and Fermented Bovine Colostrum (BCOL)

Essential amino acid contents in non-fermented and fermented BCOL are shown in Figure 2, Figure 3, Figure 4, Figure 5, Figure 6, Figure 7 and Figure 8 (concentrations of phenylalanine, valine, threonine, methionine, leucine/isoleucine, lysine, and histidine, respectively).

#### 3.2.1. Phenylalanine Content

Comparing the phenylalanine content in bovine colostrum, different tendencies were established, and in some of the BCOL samples, phenylalanine content increased after fermentation (in the A135 and D244 sample groups) (Figure 2). In contrast, in others, this content remained similar to the non-fermented samples (the C sample groups), and in others, the phenylalanine content decreased (in the B135, B244, and E244 sample groups). Furthermore, in two out of six groups of the fermented bovine colostrum, phenylalanine was not detected (in both BCOL obtained from the source of colostrum B and fermented with LUHS135 and LUHS244 LAB strains). However, the highest content of phenylalanine was found in the non-fermented B and E samples, as well as in the E samples fermented with LUHS135 strain (on average, 0.214 µmol/g) (Figure 2). Analyzed factors and their interaction were statistically significant on phenylalanine concentration in BCOL (source of colostrum *p* < 0.001, LAB strain used for fermentation *p* = 0.043, and factor interaction *p* < 0.001) (Appendix A). However, correlations between the phenylalanine concentration and LAB viable counts and pH of BCOL were not significant (Appendix A). These results can be explained by the fact that BCOL obtained from different agricultural companies can vary in their composition of other micronutrients and in the endogenous microbiota, and these factors may lead to differences in phenylalanine concentration during fermentation.

Phenylalanine is a neutral and nonpolar essential amino acid, which is an important precursor of many aromatic compounds required for normal body physiological function [54]. It was reported that, in mammals, phenylalanine can be metabolized by phenylalanine hydroxylase, and such enzymatic activities were also established in bacteria and lower eukaryote organisms [55]. Additionally, the differences in phenylalanine concentration in non-fermented bovine colostrum can be explained by the possible different effectiveness of its conversion to tyrosine [56] and, in fermented samples, the differences in phenylalanine concentration may be attained due to the distinct activities of microbial decarboxylase, which leads to different phenylalanine conversion to phenylethylamine [57].

**Figure 2 animals-13-03154-f002:** Phenylalanine concentration (µmol/g) in non-fermented and fermented bovine colostrum samples collected from different agricultural companies [A—source of colostrum in Dauksiai village, Lithuania, B—T source of colostrum in Sauselio village, Lithuania, C—source of colostrum in Geluvos village, Lithuania, D—source of colostrum in Paliepiu village, Lithuania, E—source of colostrum in Alksnupiu village, Lithuania; 135—fermented with *Lactiplantibacillus plantarum* strain 135; 244—fermented with *Lacticaseibacillus paracasei* strain 244]. —control group samples; —samples fermented with *Lp. plantarum* strain 135; —samples fermented with *Lc. paracasei* strain 244. Data were expressed as mean values ± SE. Different letters ^(a–e)^ indicate significant differences (*p* ≤ 0.05) among all treatments.

#### 3.2.2. Valine Content

Valine content in bovine colostrum is given in Figure 3. In non-fermented BCOL groups, the highest valine content was obtained in the B and E sample groups (on average, 0.566 µmol/g). However, fermentation with LUHS135 and LUHS244 strains reduced the valine content in the B samples, on average, by 82.3 and 73.2%, respectively. Moreover, a lower concentration of valine in the E sample group fermented with LUHS244 strain was observed (on average, 24.7% lower) in comparison with non-fermented samples and fermented samples with LUHS135. The factor source of colostrum was statistically significant on valine content in BCOL (*p* < 0.001) (Appendix A). However, correlations between valine concentration and the LAB viable counts and pH values were not significant (Appendix A).

**Figure 3 animals-13-03154-f003:** Valine concentration (µmol/g) in non-fermented and fermented bovine colostrum samples collected from different agricultural companies [A—source of colostrum in Dauksiai village, Lithuania, B—T source of colostrum in Sauselio village, Lithuania, C—source of colostrum in Geluvos village, Lithuania, D—source of colostrum in Paliepiu village, Lithuania, E—source of colostrum in Alksnupiu village, Lithuania; 135—fermented with *Lactiplantibacillus plantarum* strain 135; 244—fermented with *Lacticaseibacillus paracasei* strain 244]. —control group samples; —samples fermented with *Lp. plantarum* strain 135; —samples fermented with *Lc. paracasei* strain 244. Data were expressed as mean values ± SE. Different letters ^(a–f)^ indicate significant differences (*p* ≤ 0.05) among all treatments.

Valine is a branched chain aliphatic amino acid [58], a popular ingredient in preparations in human and animal nutrition [59], pharmaceuticals [60], cosmetics, antibiotics [61], anti-viral drugs [62], etc. Valine can be synthesized by microorganisms [63,64] and can be degraded through transamination pathways [65]. Additionally, it was reported that valine, in combination with leucine and isoleucine, can improve the proliferation of beneficial microbiota in vivo [66].

#### 3.2.3. Threonine Content

In comparison to non-fermented bovine colostrum groups, threonine was not detected in the A samples group; however, in other BCOL, its concentration ranged from 0.210 to 0.048 µmol/g (in B and D sample groups, respectively) (Figure 4). However, after fermentation (with both tested LAB strains), threonine was not found in the B135 and B244, as well as in the D244 samples. Opposite trends were established in the A135, C135, and D135 samples, in which the concentration of threonine increased, in comparison with non-fermented samples (on average, by 0.013 µmol/g, 1.38 and 1.65 times, respectively). Analyzed factors and their interactions were statistically significant on threonine concentration in BCOL (source of colostrum *p* = 0.006, LAB strain used for fermentation *p* = 0.017, and factor interaction *p* = 0.003) (Appendix A). However, correlations between threonine concentration with the LAB viable counts and pH of the bovine colostrum were not statistically significant (Appendix A).

Threonine is α-amino-β-hydroxybutyric acid [67], identified as an important molecule for protein synthesis, energy, and nutrient metabolism, including lipid metabolism, protein synthesis, and intestinal health and function [68,69,70]. It was reported that threonine can promote the growth of intestinal beneficial bacteria [70,71,72]. Threonine can be metabolized to glycine, acetyl CoA, and pyruvate. Nevertheless, threonine metabolism may differ depending on the physiologic state, and it is closely related to animal health and diseases [70]. Our study showed that there were significant differences (*p* ≤ 0.05) in threonine concentration in both the non-fermented and fermented BCOL groups. The obtained results could be explained either by differences in animal health status in different agricultural companies (farms) or by microbial fermentation, which can also lead to the synthesis of this amino acid [73].

**Figure 4 animals-13-03154-f004:** Threonine concentration (µmol/g) in non-fermented and fermented bovine colostrum samples collected from different agricultural companies [A—source of colostrum in Dauksiai village, Lithuania, B—T source of colostrum in Sauselio village, Lithuania, C—source of colostrum in Geluvos village, Lithuania, D—source of colostrum in Paliepiu village, Lithuania, E—source of colostrum in Alksnupiu village, Lithuania; 135—fermented with *Lactiplantibacillus plantarum* strain 135; 244—fermented with *Lacticaseibacillus paracasei* strain 244]. —control group samples; —samples fermented with *Lp. plantarum* strain 135; —samples fermented with *Lc. paracasei* strain 244. Data were expressed as mean values ± SE. Different letters ^(a–g)^ indicate significant differences (*p* ≤ 0.05) among all treatments.

#### 3.2.4. Methionine Content

Methionine concentration in bovine colostrum is shown in Figure 5. This amino acid was only detected in the non-fermented samples and E sample groups fermented with LUHS135 strain (0.012 and 0.036 µmol/g, respectively), as well as in the D samples fermented with LUHS244 strain (0.021 µmol/g). Methionine, as the principal sulfur-containing acid, is a precursor of homocysteine and glutathione [74]. It was reported that methionine could be synthesized by microorganisms [75]. On the opposite, degradation of the latter amino acid by yeasts and bacteria was also reported [76]. Our results showed that although methionine was not detected in the non-fermented E group, its presence in the E sample groups fermented with both LAB strains was established. Additionally, in fermented D samples, methionine was not found. The test of between-subject effects showed that all analyzed factors and their interaction were statistically significant on methionine concentration in BCS (*p* < 0.001) (Appendix A).

**Figure 5 animals-13-03154-f005:** Methionine concentration (µmol/g) in non-fermented and fermented bovine colostrum samples collected from different agricultural companies [A—source of colostrum in Dauksiai, Lithuania, B—T source of colostrum in Sauselio village, Lithuania, C—source of colostrum in Geluvos village, Lithuania, D—source of colostrum in Paliepiu village, Lithuania, E—source of colostrum in Alksnupiu village, Lithuania; 135—fermented with *Lactiplantibacillus plantarum* strain 135; 244—fermented with *Lacticaseibacillus paracasei* strain 244]. —control group samples; —samples fermented with *Lp. plantarum* strain 135; —samples fermented with *Lc. paracasei* strain 244. Data were expressed as mean values ± SE. Different letters ^(a–c)^ indicate significant differences (*p* ≤ 0.05) among all treatments.

#### 3.2.5. Leucine and Isoleucine Content

Leucine and isoleucine were found in all bovine colostrum groups, and the highest content of these amino acids was determined in the non-fermented B and E groups, as well as in the E sample groups fermented with LUHS135 (on average, 0.808 µmol/g) (Figure 6). Comparing the BCOL samples before and after fermentation, a lower concentration of leucine and isoleucine was unveiled in the B135, A244, B244, and E244 samples when compared with the non-fermented samples: on average, 4.49, 1.46, 5.35, and 1.68 times lower, respectively. The source of colostrum was a significant factor concerning leucine and isoleucine concentration in BCOL (*p* = 0.013) (Appendix A). However, the correlations between these amino acid concentrations and the LAB viable counts and pH of the BCOL were not statistically significant (Appendix A).

Leucine is a unique branched-chain amino acid that plays an important role as a nutrient sensor [77,78,79]. The branched-chain amino acids are involved in skeletal muscle composition [80]. Isoleucine stimulates glucose uptake in the muscle and whole body glucose oxidation, in addition to depressing gluconeogenesis in the liver of in vivo models with rats [81]. Leucine can be assimilated by bacteria when saccharides or other carbon sources in the habitat are depleted [82]. In contrast, branched-chain amino acids, including leucine and isoleucine, can be produced and excreted by various bacteria strains, including pathogenic microorganisms [83].

**Figure 6 animals-13-03154-f006:** Leucine and isoleucine concentration (µmol/g) in non-fermented and fermented bovine colostrum samples collected from different agricultural companies [A—source of colostrum in Dauksiai village, Lithuania, B—T source of colostrum in Sauselio village, Lithuania, C—source of colostrum in Geluvos village, Lithuania, D—source of colostrum in Paliepiu village, Lithuania, E—source of colostrum in Alksnupiu village, Lithuania; 135—fermented with *Lactiplantibacillus plantarum* strain 135; 244—fermented with *Lacticaseibacillus paracasei* strain 244]. —control group samples; —samples fermented with *Lp. plantarum* strain 135; —samples fermented with *Lc. paracasei* strain 244. Data were expressed as mean values ± SE. Different letters ^(a–f)^ indicate significant differences (*p* ≤ 0.05) among all treatments.

#### 3.2.6. Lysine Content

Lysine was not detected in the non-fermented and fermented B sample groups. However, the highest concentration of lysine was revealed in the non-fermented bovine colostrum samples and E samples fermented with LUHS135 (on average, 0.120 µmol/g) (Figure 7). In contrast, in the non-fermented and fermented sample groups, lysine content was lower in the A135, C135, C244, and E244 samples (on average, by 2.1, 1.35, 1.62, and 2.64 times, respectively, in comparison with the non-fermented samples). However, opposite tendencies were found in the A samples group, in which, after fermentation with the LUHS244 strain, lysine concentration increased by 38.1%, on average.

Lysine is often the first-limiting amino acid in the diets of mammals [84]. The lysine degradation mechanism by bacteria was reported by Sabo et al. [85]. *Escherichia coli*, as a microorganism possessing lysine decarboxylase activity, was also described [85,86]. However, lysine synthesis through the bacterial (e.g., *Corynebacterium glutamicum*) fermentation of carbohydrates is the most predominant method for the production of this amino acid [87]. Our study showed that the analyzed factors and their interaction were statistically significant on lysine concentration in BCOL (*p* < 0.001) (Appendix A). However, correlations between lysine concentration and the LAB viable counts and pH of BCOL were not statistically significant (Appendix A).

**Figure 7 animals-13-03154-f007:** Lysine concentration (µmol/g) in non-fermented and fermented bovine colostrum samples collected from different agricultural companies [A—source of colostrum in Dauksiai village, Lithuania, B—T source of colostrum in Sauselio village, Lithuania, C—source of colostrum in Geluvos village, Lithuania, D—source of colostrum in Paliepiu village, Lithuania, E—source of colostrum in Alksnupiu village, Lithuania; 135—fermented with *Lactiplantibacillus plantarum* strain 135; 244—fermented with *Lacticaseibacillus paracasei* strain 244]. —control group samples; —samples fermented with *Lp. plantarum* strain 135; —samples fermented with *Lc. paracasei* strain 244. Data were expressed as mean values ± SE. Different letters ^(a–f)^ indicate significant differences (*p* ≤ 0.05) among all treatments.

#### 3.2.7. Histidine Content

Histidine was not detected in the non-fermented A and C sample groups, as well as in the A, B, and C samples fermented with the LUHS135 strain and the A and B samples fermented with the LUHS244 strain (Figure 8). Although in non-fermented and C bovine colostrum sample groups fermented with LUHS135 strain, the histidine was not found, the samples fermented with LUHS244 strain showed, on average, a histidine content of 0.019 µmol/g. Moreover, fermentation with the LUHS135 strain increased histidine concentration in the D samples group (on average, by 18.9%) in comparison with the non-fermented ones. Analyzed factors and their interaction were statistically significant on histidine concentration in BCOL (source of colostrum *p* < 0.001, LAB strain used for fermentation *p* = 0.016, and factor interaction *p* < 0.001) (Appendix A). However, correlations between histidine concentration and the LAB viable counts and pH of BCOL were not statistically significant (Appendix A).

Histidine is an essential amino acid and must be obtained through the diet [88,89], playing an important role in nitrogen balance [90]. It was reported that histidine is indispensable for ruminants [91], and histidine supplementation induces an increase in milk yield and protein concentration in milk [90]. However, histidine can be converted to histamine by histidine decarboxylase [92]. Additionally to desirable microorganisms (e.g., LAB), contamination of the substrate with non-desirable microbiota can also lead to higher histamine concentration [93]. Therefore, histamine and histidine concentration in the substrate can vary because of both the primary microbiological contamination of the substrate and the technological strains employed for the fermentation.

**Figure 8 animals-13-03154-f008:** Histidine concentration (µmol/g) in non-fermented and fermented bovine colostrum samples collected from different agricultural companies [A—source of colostrum in Dauksiai village, Lithuania, B—T source of colostrum in Sauselio village, Lithuania, C—source of colostrum in Geluvos village, Lithuania, D—source of colostrum in Paliepiu village, Lithuania, E—source of colostrum in Alksnupiu village, Lithuania; 135—fermented *with Lactiplantibacillus plantarum* strain; 244—fermented with *Lacticaseibacillus paracasei* strain]. —control group samples; —samples fermented with *Lp. plantarum* strain 135; —samples fermented with *Lc. paracasei* strain 244. Data were expressed as mean values ± SE. Different letters ^(a–e)^ indicate significant differences (*p* ≤ 0.05) among all treatments.

#### 3.2.8. Alanine Content

Non-essential amino acids and GABA concentrations in non-fermented and fermented bovine colostrum are shown in Figure 9, Figure 10, Figure 11, Figure 12, Figure 13, Figure 14, Figure 15, Figure 16, Figure 17 and Figure 18 (concentrations of alanine, arginine, aspartic acid, glutamic acid, glutamine, glycine, proline, serine, tyrosine, and GABA, respectively).

The concentration of alanine in BCOL is depicted in Figure 9. In comparison to non-fermented samples, the highest concentration of alanine was found in the B samples group (1.92 µmol/g). Samples A, C, D, and E showed, respectively, on average, 61.4, 52.1, 55.6, and 21.4% lower alanine concentrations than the B sample groups. Contrasting alanine concentration in non-fermented against fermented samples, the concentration in the samples B135, C135, B244, C244, and E244 was lower; however, in the A135 and E135 sample groups, alanine concentration after fermentation was, respectively, on average, 52.3 and 25.8% higher than in the non-fermented samples.

Alanine has extensive applications [63,94,95,96,97,98,99] and can be synthesized by various microorganisms [94,100,101,102,103,103,104,105]. However, some of the LAB strains can metabolize alanine to volatile compounds, and these reactions are initiated mainly by oxidative deamination and decarboxylation [106]. Additionally, the patterns of amino acid metabolites depend upon a qualitative and quantitative balance of the amino acids, the pH of the substrate, and the processing temperature [106]. Our study indicated that the analyzed factors and their interaction were statistically significant on alanine concentration in BCOL (source of colostrum and LAB strain used for fermentation *p* < 0.001, and factor interaction *p* = 0.033) (Appendix A). Moreover, alanine showed a positive moderate correlation with BCOL pH values (r = 0.409, *p* = 0.005) (Appendix A).

**Figure 9 animals-13-03154-f009:** Alanine concentration (µmol/g) in non-fermented and fermented bovine colostrum samples collected from different agricultural companies [A—source of colostrum in Dauksiai, Lithuania, B—T source of colostrum in Sauselio village, Lithuania, C—source of colostrum in Geluvos village, Lithuania, D—source of colostrum in Paliepiu village, Lithuania, E—source of colostrum in Alksnupiu village, Lithuania; 135—fermented with *Lactiplantibacillus plantarum* strain 135; 244—fermented with *Lacticaseibacillus paracasei* strain 244]. —control group samples; —samples fermented with *Lp. plantarum* strain 135; —samples fermented with *Lc. paracasei* strain 244. Data were expressed as mean values ± SE. Different letters ^(a–f)^ indicate significant differences (*p* ≤ 0.05) among all treatments.

#### 3.2.9. Arginine Content

Arginine was not detected in the B135 samples group, and the highest arginine content was found in the non-fermented C samples (0.187 µmol/g) (Figure 10). Comparing arginine content in non-fermented and fermented BCOL, fermentation with LUHS135 reduced arginine content in the C135 sample group, and fermentation with LUHS244 reduced arginine content in the B244 and C244 sample groups (on average, by 31.0, 51.6 and 51.9%, respectively). Reversely, higher arginine content was found in the D135, E135, A244, and D244 sample groups than in the non-fermented ones, on average, respectively, it was 1.35, 1.29, 1.91, and 1.25 times higher.

Arginine can be metabolized by arginine decarboxylase, glycine amidinotransferase, arginase, and nitric oxide synthase [107,108]. It was reported that the *Furfurilactobacillus rossiae* D87 strain, isolated from sourdough, can metabolize arginine via the decarboxylase pathway to ornithine as well as to putrescine [109]. Additionally, L-arginine is involved in the synthesis of proteins, urea, creatine, proline, nitric oxide, and BAs (putrescine, spermine, and spermidine) [110]. Biogenic amines are one of the end products of L-arginine metabolism, which possesses various regulatory functions in the body [111,112,113,114,115,116,117,118,119,120,121,122,123]. It was reported that arginine is an important nutrient among the amino acids during lacto-fermentation [124]. Our study showed that the source of colostrum and factor interaction were statistically significant in arginine content in BCOL (*p* < 0.001) (Appendix A). However, correlations between arginine concentration with LAB viable counts and pH of BCOL were not statistically significant (Appendix A).

**Figure 10 animals-13-03154-f010:** Arginine concentration (µmol/g) in non-fermented and fermented bovine colostrum samples collected from different agricultural companies [A—source of colostrum in Dauksiai village, Lithuania, B—T source of colostrum in Sauselio village, Lithuania, C—source of colostrum in Geluvos village, Lithuania, D—source of colostrum in Paliepiu village, Lithuania, E—source of colostrum in Alksnupiu village, Lithuania; 135—fermented with *Lactiplantibacillus plantarum* strain 135; 244—fermented with *Lacticaseibacillus paracasei* strain 244]. —control group samples; —samples fermented with *Lp. plantarum* strain 135; —samples fermented with *Lc. paracasei* strain 244. Different letters ^(a–g)^ indicate significant differences (*p* ≤ 0.05) among all treatments.

#### 3.2.10. Aspartic Acid Content

Aspartic acid was not detected in the A, C, B135, A244, and B244 bovine colostrum sample groups (Figure 11). Despite the fact that aspartic acid was not found in the non-fermented A and C sample groups, its content in the A and C samples fermented with LUHS135 strain was, on average, 0.065 and 0.034 µmol/g, respectively. However, in the A sample group fermented with the LUHS244 strain, aspartic acid was not detected, in contrast to the C244 samples, in which aspartic acid content was, on average, 0.044 µmol/g. Comparing the non-fermented and fermented sample groups, fermentation decreased aspartic acid content in the B135, D135, E135, B244, and E244 samples, and in contrast, fermentation increased aspartic acid content in the A135, C135, and C244 samples.

**Figure 11 animals-13-03154-f011:** Aspartic acid concentration (µmol/g) in non-fermented and fermented bovine colostrum samples collected from different agricultural companies [A—source of colostrum in Dauksiai, Lithuania, B—T source of colostrum in Sauselio village, Lithuania, C—source of colostrum in Geluvos village, Lithuania, D—source of colostrum in Paliepiu village, Lithuania, E—source of colostrum in Alksnupiu village, Lithuania; 135—fermented with *Lactiplantibacillus plantarum* strain 135; 244—fermented with *Lacticaseibacillus paracasei* strain 244]. —control group samples; —samples fermented with *Lp. plantarum* strain 135; —samples fermented with *Lc. paracasei* strain 244. Data were expressed as mean values ± SE. Different letters ^(a–f)^ indicate significant differences (*p* ≤ 0.05) among all treatments.

Aspartic acid is a four-carbon amino acid produced in L and D-isoforms. The latter amino acid has a wide application in the food, beverage, pharmaceutical, cosmetic, and agriculture industries [125,126]. Aspartic acid can be synthesized during the fermentation process by enzymatic conversion [127,128]. Several bacteria species (*Pseudomonas* spp., *Bacillus* spp., and *Proteus* spp.) were identified as very good producers of aspartic acid [129]. Moreover, it was reported that aspartic acid is susceptible to biological degradation [130]. Our study showed that the analyzed factors and their interactions were statistically significant on aspartic acid concentration in BCOL (*p* < 0.001) (Appendix A). Conversely, correlations between aspartic acid concentration and the LAB viable counts and pH of BCOL were not statistically significant (Appendix A).

#### 3.2.11. Glutamic Acid Content

Contrasting glutamic acid concentration in non-fermented bovine colostrum sample groups, the highest content was found in the E samples (0.678 µmol/g). Chiefly, the A, B, C, and D samples showed, on average, 46.3, 34.7, 42.5, and 44.0% lower glutamic acid concentrations (Figure 12). When comparing non-fermented and fermented samples, the fermentation with LUHS135 strain increased glutamic acid concentration in the A135 and E135 sample groups (on average, by 28.6 and 25.2%, respectively), and the fermentation with LUHS244 increased glutamic acid concentration in the D244 sample group (on average, by 32.6%). The analyzed factor source of colostrum was statistically significant (*p* = 0.004) on glutamic acid concentration in BCOL (Appendix A), and a positive weak correlation was found between glutamic acid concentration and LAB viable counts in BCOL (r = 0.296, *p* = 0.049) (Appendix A).

It was described that glutamic acid can be synthesized by microbial racemization, polymerization, transfer, and catabolism, and *Bacillus* species are the most widely studied producing strains [131]. In addition to bacteria, archaea and eukaryotes are also good glutamic acid producers [132,133,134,135]. However, glutamic acid production is strain and substrate-specific dependent. For instance, *Corynebacterium* efficiently synthesizes glutamic acid at high temperatures, while *Pantoea ananatis* produces glutamic acid under acidic conditions [136].

**Figure 12 animals-13-03154-f012:** Glutamic acid concentration (µmol/g) in non-fermented and fermented bovine colostrum samples collected from different agricultural companies [A—source of colostrum in Dauksiai, Lithuania, B—T source of colostrum in Sauselio village, Lithuania, C—source of colostrum in Geluvos village, Lithuania, D—source of colostrum in Paliepiu village, Lithuania, E—source of colostrum in Alksnupiu village, Lithuania; 135—fermented with *Lactiplantibacillus plantarum* strain 135; 244—fermented with *Lacticaseibacillus paracasei* strain 244]. —control group samples; —samples fermented with *Lp. plantarum* strain 135; —samples fermented with *Lc. paracasei* strain 244. Data were expressed as mean values ± SE. Different letters ^(a–f)^ indicate significant differences (*p* ≤ 0.05) among all treatments.

#### 3.2.12. Glutamine Content

The highest glutamine concentration was found in the non-fermented bovine colostrum A sample group (1.43 µmol/g) (Figure 13). Other non-fermented samples (B, C, D, and E) showed, on average, 2.82, 2.64, 1.69, and 2.27 times lower glutamine content than the A sample group. Fermentation increased glutamine concentration in the E135 and C244 samples (on average, by 47.3 and 73.6%, respectively) but decreased in sample groups A135, A244, B244, and D244 (on average, by 42.4, 64.8, 30.6 and 20.8%, respectively). Analyzed factors and their interaction were not statistically significant on glutamine content in BCOL (Appendix A), and correlations between glutamine concentration and LAB viable counts and pH of BCOL were not statistically significant (Appendix A).

In addition to its participation in nitrogen metabolism, glutamine is known as a precursor of other amino acids, purines, pyrimidines, pyridine coenzymes, and complex carbohydrates [137]. The latter amino acids can be synthesized by direct fermentation with certain bacterial strains [138,139]. However, it was reported that ruminants can rapidly break down glutamine into asparagine and nicotinamide, the latter of which can lead to exceeding ammonia production and turn back to glutamine synthesis [140]. Our research showed that the analyzed factors and their interaction were not statistically significant on glutamine content in BCOL (Appendix A), as well as the correlations between glutamine concentration and LAB viable counts and pH of BCOL (Appendix A). These findings can be explained by the possible glutamine cycle, as comprehensively explained by Warner et al. [140].

**Figure 13 animals-13-03154-f013:** Glutamine concentration (µmol/g) in non-fermented and fermented bovine colostrum samples collected from different agricultural companies [A—source of colostrum in Dauksiai village, Lithuania, B—T source of colostrum in Sauselio village, Lithuania, C—source of colostrum in Geluvos village, Lithuania, D—source of colostrum in Paliepiu village, Lithuania, E—source of colostrum in Alksnupiu village, Lithuania; 135—fermented with *Lactiplantibacillus plantarum* strain 135; 244 –fermented with *Lacticaseibacillus paracasei* strain 244]. —control group samples; —samples fermented with *Lp. plantarum* strain 135; —samples fermented with *Lc. paracasei* strain 244. Data were expressed as mean values ± SE. Different letters ^(a–e)^ indicate significant differences (*p* ≤ 0.05) among all treatments.

#### 3.2.13. Glycine Content

Comparing the glycine concentration in the non-fermented bovine colostrum, the highest content was obtained in the B samples group (on average, 2.47 µmol/g) (Figure 14). Sample groups A, C, D, and E showed, on average, 77.6, 59.6, 64.4, and 19.8% lower glycine concentration than the B samples group. When comparing non-fermented and fermented sample groups, fermentation increased glycine concentration in the sample groups A135, E135, and D135 (on average, by 97.0, 30.3, and 52.5%, respectively). However, lower glycine concentration was found in the fermented sample groups B135, C135, B244, C244, and E244 (on average, by 21.3, 1.75, 5.24, 1.62, and 1.94 times, respectively).

Glycine is a simple amino acid that can be degraded through the glycine cleavage system, serine hydroxymethyltransferase, and conversion to glyoxylate by peroxisomal D-amino acid oxidase [141]. The main products of glycine metabolism are ammonia and CO_2_ [142]. Some of the bacteria species can utilize glycine as the sole carbon and nitrogen source through the glycine dehydrogenase pathway. Moreover, it was reported that the effect of substrate constituent concentrations on glycine synthesis is quite complex [143]. Our study showed that the analyzed factors and their interaction were statistically significant on glycine concentration in BCOL (source of colostrum *p* = 0.012, LAB strain used for fermentation *p* = 0.001, and factors interaction *p* = 0.001) (Appendix A). Glycine showed a positive moderate correlation with the pH values of BCOL (r = 0.422, *p* = 0.004) (Appendix A).

**Figure 14 animals-13-03154-f014:** Glycine concentration (µmol/g) in non-fermented and fermented bovine colostrum samples collected from different agricultural companies [A—source of colostrum in Dauksiai village, Lithuania, B—T source of colostrum in Sauselio village, Lithuania, C—source of colostrum in Geluvos village, Lithuania, D—source of colostrum in Paliepiu village, Lithuania, E—source of colostrum in Alksnupiu village, Lithuania; 135—fermented with *Lactiplantibacillus plantarum* strain 135; 244—fermented with *Lacticaseibacillus paracasei* strain 244]. —control group samples; —samples fermented with *Lp. plantarum* strain 135; —samples fermented with *Lc. paracasei* strain 244. Data were expressed as mean values ± SE. Different letters ^(a–g)^ indicate significant differences (*p* ≤ 0.05) among all treatments.

#### 3.2.14. Proline Content

Proline concentration in bovine colostrum is given in Figure 15. Regarding the non-fermented groups, the highest concentration of proline was found in the B group (0.641 µmol/g). Proline concentration in samples C, D, and E was, on average, 0.433 µmol/g, whereas in samples A group was, on average, 0.241 µmol/g. When comparing non-fermented and fermented sample groups, fermentation increased proline content in sample groups A135, D135, and E135 (on average, by 34.9, 15.3 and 43.0%, respectively). However, fermented sample groups B244 and B135 showed, on average, 62.3 and 67.7% lower proline content. The interaction of analyzed factors was statistically significant on proline concentration in BCOL (*p* = 0.006) (Appendix A). However, correlations between proline concentration with the LAB viable counts and pH of BCOL were not statistically significant (Appendix A).

It was reported that correlations exist between plasma proline levels and the microbiota composition of the persons [144]. Higher levels of proline were associated with higher numbers of *Parabacteroides* and *Acidaminococcus* species, and lower numbers were associated with *Bifidobacterium*. Moreover, proline can be accumulated during food fermentation processes because the yeast *Saccharomyces cerevisiae* poorly assimilates proline, while proline can be utilized by *Zygoascus*, *Galactomyces*, and *Magnusiomyces* genera [145]. Despite the ability of mammalian cells to synthesize proline, dietary sources of proline are thought to be essential to maintaining a healthy function [146].

**Figure 15 animals-13-03154-f015:** Proline concentration (µmol/g) in non-fermented and fermented bovine colostrum samples collected from different agricultural companies [A—source of colostrum in Dauksiai village, Lithuania, B—T source of colostrum in Sauselio village, Lithuania, C—source of colostrum in Geluvos village, Lithuania, D—source of colostrum in Paliepiu village, Lithuania, E—source of colostrum in Alksnupiu village, Lithuania; 135—fermented with *Lactiplantibacillus plantarum* strain 135; 244—fermented with *Lacticaseibacillus paracasei* strain 244]. —control group samples; —samples fermented with *Lp. plantarum* strain 135; —samples fermented with *Lc. paracasei* strain 244. Data were expressed as mean values ± SE. Different letters ^(a–e)^ indicate significant differences (*p* ≤ 0.05) among all treatments.

#### 3.2.15. Serine Content

Serine was not detected in the non-fermented bovine colostrum sample groups C and D, as well as in the sample groups B135 and C135 fermented with LUHS135, and in the sample group C244 fermented with LUHS244 (Figure 16). When observing non-fermented and fermented sample groups, despite that serine was not detected in the non-fermented D samples, in the samples fermented with LUHS135 and LUHS244 strains, the content was, on average, 0.010 and 0.109 µmol/g, respectively. Moreover, fermentation with the LUHS135 strain increased serine concentration in the A135 samples (on average, by 2.74 times). The source of colostrum and interaction of analyzed factors (source of colostrum and LAB strain used for fermentation) were statistically significant on serine concentration in BCOL (*p* < 0.001) (Appendix A). However, correlations between serine concentration with the LAB viable counts and pH of BCOL were not statistically significant (Appendix A).

**Figure 16 animals-13-03154-f016:** Serine concentration (µmol/g) in non-fermented and fermented bovine colostrum samples collected from different agricultural companies [A—source of colostrum in Dauksiai village, Lithuania, B—T source of colostrum in Sauselio village, Lithuania, C—source of colostrum in Geluvos village, Lithuania, D—source of colostrum in Paliepiu village, Lithuania, E—source of colostrum in Alksnupiu village, Lithuania; 135—fermented with *Lactiplantibacillus plantarum* strain 135; 244—fermented with *Lacticaseibacillus paracasei* strain 244]. —control group samples; —samples fermented with *Lp. plantarum* strain 135; —samples fermented with *Lc. paracasei* strain 244. Data were expressed as mean values ± SE. Different letters ^(a–g)^ indicate significant differences (*p* ≤ 0.05) among all treatments.

In mammals, L-serine can be synthesized from 3-phosphoglycerate (3-PG) and glycine [147]. However, in the diet, L-serine content is, on average, 3.5% of the total protein content [148]. Additionally, serine can be converted by serine-pyruvate transaminase, glycerate dehydrogenase, and glycerate kinase to 2-phosphoglycerate [149]. However, serine can be synthesized by fermentation of glycerol and glucose-rich substrates with *E. coli* [150] and other strains [151,152].

#### 3.2.16. Tyrosine Content

Tyrosine concentration in bovine colostrum is shown in Figure 17. In comparison to non-fermented sample groups, the highest tyrosine content was found in the C and E group samples (on average, 0.109 µmol/g). Samples A, B, and D showed, respectively, on average, 1.54, 2.79, and 1.95 times lower tyrosine content than in the C and E sample groups. Comparing the non-fermented and fermented sample groups, fermentation increased tyrosine content in the C135, A244, and C244 samples (on average, by 42.9, 52.1 and 25.7%, respectively) and decreased in the A135, D135, B244, and D244 sample groups (on average, by 49.3, 41.1, 56.4 and 30.4%, respectively). All analyzed factors and their interaction were statistically significant on tyrosine concentration in BCOL (*p* < 0.001) (Appendix A). However, correlations between tyrosine concentration and LAB viable counts and pH of BCOL were not statistically significant (Appendix A).

Tyrosine is used as a nutritional supplement for humans and animals. It is the precursor of dopamine and thyroxine. Tyrosine can be synthesized during microbial fermentations [153]. However, the concentration of this amino acid depends on the strain used for fermentation and the parameters of fermentation [153]. Our study showed that all analyzed factors and their interaction were statistically significant on tyrosine concentration in BCOL (*p* < 0.001) (Appendix A). However, correlations between tyrosine concentration and the LAB viable counts and pH of BCOL were not statistically significant (Appendix A).

**Figure 17 animals-13-03154-f017:** Tyrosine concentration (µmol/g) in non-fermented and fermented bovine colostrum samples collected from different agricultural companies [A—source of colostrum in Dauksiai, Lithuania, B—T source of colostrum in Sauselio village, Lithuania, C—source of colostrum in Geluvos village, Lithuania, D—source of colostrum in Paliepiu village, Lithuania, E—source of colostrum in Alksnupiu village, Lithuania; 135—fermented with *Lactiplantibacillus plantarum* strain 135; 244—fermented with *Lacticaseibacillus paracasei* strain 244]. —control group samples; —samples fermented with *Lp. plantarum* strain 135; —samples fermented with *Lc. paracasei* strain 244. Data were expressed as mean values ± SE. Different letters ^(a–f)^ indicate significant differences (*p* ≤ 0.05) among all treatments.

#### 3.2.17. Gamma Amino-Butyric Acid (GABA) Content

Gamma amino-butyric acid (GABA) concentration in non-fermented and fermented bovine colostrum is represented in Figure 18. The highest GABA content was found in the non-fermented B group samples (0.408 µmol/g). When comparing the non-fermented and fermented sample groups, we found out that in 5 out of 10 fermented sample groups, GABA concentration was higher than in non-fermented samples (in D135, E135, C244, D244, and E244, the concentration was, on average, 1.37, 1.62, 1.35, 1.65 and 1.82 times higher, respectively). Contrarily, fermentation reduced GABA content in the A135, B135, and B244 sample groups, and in sample groups C135 and A244, the concentration of GABA was similar to the non-fermented samples.

The beneficial effects of GABA are widely described [154,155,156,157,158,159]. Additionally, in microorganisms, GABA is involved in spore germination and causes acid resistance in bacteria [160]. The synthesis of GABA is catalyzed by the glutamic acid decarboxylase and the cofactor pyridoxal-5-phosphate from L-glutamate [161,162]. GABA has been isolated from many sources, including, LAB strains (e.g., *Streptococcus thermophilus*, *Levilactobacillus brevis*, *Lacticaseibacillus paracasei*, *Lactobacillus futsaii*, *Lactiplantibacillus plantarum,* and *Bifidobacterium adolescentis*), yeasts (e.g., *Saccharomyces cerevisiae, Kluyveromyces marxianus, Kazachstania unispora, Sporobolomyces carnicolor, Sporobolomyces ruberrimus, Nakazawaea holstii,* and *Pichia scolyti*), and molds (e.g., *Aspergillus oryzae*) [161,162,163,163,164,165,166,167,168,169,170,171,172,173,174,175,176,177,178,179,180,181,182]. GABA is also found in fermented foods [168,170,174,177,183,184,185]. GABA production by microorganisms, as a safe and eco-friendly process, has been extensively studied [186]. In contrast, GABA may be degraded to succinate by GABA amino-transferase and semialdehyde dehydrogenase [186]. However, many factors influence GABA synthesis, including temperature, pH, substrate and culture characteristics, duration of fermentation, etc. [161,184,187,188]. In relation to these factors, GABA concentrations in the substrate can vary. It was reported that *Saccharomyces cerevisiae* may consume GABA, thereby reducing the amount of the produced GABA [176]. Our study showed that the LAB strains used for fermentation and factor interaction were statistically significant on GABA concentration in BCOL (*p* < 0.001) (Appendix A). However, correlations between GABA concentration and LAB viable counts and pH of BCOL were not statistically significant (Appendix A).

**Figure 18 animals-13-03154-f018:** Gamma amino-butyric acid (GABA) concentration (µmol/g) in non-fermented and fermented bovine colostrum samples collected from different agricultural companies [A—source of colostrum in Dauksiai, Lithuania, B—T source of colostrum in Sauselio village, Lithuania, C—source of colostrum in Geluvos village, Lithuania, D—source of colostrum in Paliepiu village, Lithuania, E—source of colostrum in Alksnupiu village, Lithuania; 135—fermented with *Lactiplantibacillus plantarum* strain 135; 244—fermented with *Lacticaseibacillus paracasei* strain 244]. —control group samples; —samples fermented with *Lp. plantarum* strain 135; —samples fermented with *Lc. paracasei* strain 244. Data were expressed as mean values ± SE. Different letters ^(a–f)^ indicate significant differences (*p* ≤ 0.05) among all treatments.

Finally, our results revealed statistically significant differences in amino acids between the BCOL obtained from different agricultural companies, and these differences, in most cases, led to distinct results in amino acid content after fermentation. Individual amino acids may possess different physiological activities. Therefore, products possessing various functional properties could be created because of the different properties and composition of the fermented bovine colostrum. This could provide a broader spectrum of bovine colostrum applications in the food, nutraceutical, pharmaceutical, and cosmetic industries, among others. Moreover, to ensure the stability of the composition of the raw material (BCOL), constant feeding, keeping conditions, and breed selection should be applied as far as possible.

### 3.3. Biogenic Amine (BA) Formation in Bovine Colostrum (BCOL) during the Fermentation Process

Individual BA concentrations in bovine colostrum are presented in Figure 19a, whereas total BA contents in bovine colostrum are shown in Figure 19b. For the comparison of the formation of individual BA in BCOL, it should be noted that spermine and spermidine were not detected in BCOL, and histamine was established just in one out of 15 analyzed sample groups (in non-fermented E samples, histamine content was on average, 16.8 mg/kg). Conversely, tyramine was formed in most of the samples, except in non-fermented E and in E135 and E244 fermented with LUHS135 and LUHS244. The highest content of tyramine was found in the C135 sample group fermented with LUHS135 (26.5 mg/kg). Tyramine showed weak negative correlations with aspartic amino acid and histidine (r = −0.369, *p* = 0.013 and r = −0.367, *p* = 0.013, respectively), as well as a positive weak correlation with alanine content (r = 0.324, *p* = 0.030) (Appendix A). A statistically significant factor affecting the tyramine formation was the source of colostrum (*p* = 0.002) (Appendix A). Tryptamine was formed in 9 out of the 15 analyzed samples, and its content ranged from null (in sample groups A, C, E135, A244, B244, and C244) to 7.34 mg/kg (in the D135 sample group). All analyzed factors and their interaction were statistically significant regarding the tryptamine formation in BCOL (*p* < 0.001) (Appendix A). Furthermore, tryptamine showed negative weak correlations with the amino acid alanine and lysine (r = −0.337, *p* = 0.024 and r = −0.306, *p* = 0.041, respectively), as well as negative moderate correlation between tryptamine and tyrosine (r = −0.568, *p* < 0.001) (Appendix A). Phenylethylamine was found in all analyzed bovine colostrum sample groups, and its content in all groups did not exceed 11 mg/kg. All analyzed factors and their interaction were statistically significant in phenylethylamine content in BCOL (*p* < 0.001) (Appendix A). Additionally, phenylethylamine content in BCOL showed a negative, weak correlation with lysine (r = −0.376, *p* = 0.011) (Appendix A). Putrescine and cadaverine were the main BAs in BCOL. The highest putrescine content was found in the non-fermented C sample group (160.6 mg/kg), and the lowest concentration of putrescine was noticed in the non-fermented E sample group (38.1 mg/kg). Putrescine content in BCOL showed significant correlations with arginine, glutamine, glutamic acid, serine, aspartic acid, glycine, alanine, methionine, phenylalanine, and histidine (Appendix A). Moreover, all analyzed factors and their interaction were significant for putrescine formation in BCOL (Appendix A). The lowest cadaverine content was found in the non-fermented E sample group and in D244 samples fermented with LUHS244 (on average, 34.5 mg/kg). The highest cadaverine content was found in the C135 samples fermented with the LUHS135 strain (140.4 mg/kg). Cadaverine content in BCOL showed significant correlations with glutamic acid, serine, aspartic acid, valine, phenylalanine, leucine, isoleucine, lysine, and histidine (Appendix A). Additionally, all analyzed factors and their interaction were statistically significant for cadaverine formation in BCOL (Appendix A). It should be mentioned that in this study, statistically significant correlations between individual BA and LAB viable counts and pH values of BCOL were not detected (Appendix A). Moreover, putrescine and cadaverine showed negative, weak correlations with GABA content in BCOL (r = −0.307, *p* = 0.040 and r = −0.361, *p* = 0.015, respectively). Comparing the total content of BA in non-fermented and fermented sample groups, fermentation increased total BA content in the A135, A244, B135, and B244 sample groups (on average, by 85.7, 84.5, 48.4 and 47.0%, respectively) (Figure 19b). However, opposite tendencies were found in the D135, D244, E135, and E244 sample groups, in which the total BA content was lower than in non-fermented samples (on average, 14.8, 27.7, 21.8 and 49.5% lower, respectively). Total BA content showed statistically significant correlations with glutamic acid, serine, aspartic acid, valine, methionine, phenylalanine, histidine, and GABA content (Appendix A). On the other hand, the source of colostrum was statistically a significant factor for total BA (Appendix A).

Microbial decarboxylation of amino acids leads to tyramine, histamine, putrescine, and cadaverine formation [189,190]. Moreover, putrescine and cadaverine can be formed from the amino acids ornithine and lysine, respectively [191]. Tyramine is associated with non-desirable effects on the vascular system, and histamine is known as a vasodilator [192]. Additionally, it was reported that the sum of primary, secondary, and tertiary BAs must be taken into consideration because tyramine causes migraines, and putrescine and cadaverine potentiate intoxication in the presence of other BAs [193,194]. Additionally, the differences in amino acid concentration led to different BA formation in different bovine colostrum groups during the lacto-fermentation. Additionally, correlations were found between the total BA content and glutamic acid, serine, aspartic acid, valine, methionine, phenylalanine, histidine, and GABA content (Appendix A). For the total BA content, the source of colostrum proved to be a statistically significant factor (Appendix A). The changes in BA formation during BCOL fermentation may be related to the fact that different bacteria strains can possess different characteristics (e.g., decarboxylase activity, proteolytic activity, etc.) for the amino acid conversion to BAs [57]. These activities and production of BA highly depend on environmental conditions (pH and temperature) and are also substrate-specific related. Finally, it can be stated that the primary microbiota (both desirable and non-desirable) of bovine colostrum can also take part in the BA formation.

**Figure 19 animals-13-03154-f019:** Individual biogenic amine (BA) contents (**a**) and total BA content (**b**) in non-fermented and fermented bovine colostrum sample (BCOL) groups [A—source of colostrum in Dauksiai, Lithuania, B—T source of colostrum in Sauselio village, Lithuania, C—source of colostrum in Geluvos village, Lithuania, D—source of colostrum in Paliepiu village, Lithuania, E—source of colostrum in Alksnupiu village, Lithuania; 135—*Lactiplantibacillus plantarum* strain 135; 244—*Lacticaseibacillus paracasei* strain 244; TRP—tryptamine; PHE—phenylethylamine; PUT—putrescine; CAD—cadaverine; HIS—histamine; TYR—tyramine; SPRMD—spermidine; SPRM—spermine; BA—biogenic amines]. Data were expressed as mean values ± SE.]. —control group samples; —samples fermented with *Lp. plantarum* strain 135; —samples fermented with *Lc. paracasei* strain 244. For each biogenic amine, different letters ^(a–j)^ indicate a significant difference (*p* ≤ 0.05) (Figure 19a) among all treatments. Different letters ^(a–f)^ indicate a significant difference (*p* ≤ 0.05) (Figure 19b) among all treatments.

### 3.4. Influence of Fermentation on Fatty Acid (FA) Profile in Bovine Colostrum (BCOL) Samples

Individual fatty acid (FA) content of the bovine colostrum is presented in Figure 20a. Compared to the butyric acid content (C4:0) in the non-fermented samples, it was found that the content ranged from 1.66 to 2.27% of the total fat content (in the A and B sample groups, respectively). In the sample groups fermented with the LUHS135 strain, its content ranged from 1.37 to 2.40% of the total fat content (in the A135 and E135 sample groups, respectively), and in the sample groups fermented with the LUHS244 strain, its content ranged between 1.34 and 2.37% of the total fat content (in the C244 and D244 sample groups, respectively). Analyzed factors and their interaction were not statistically significant on C4:0 content in BCOL (Appendix A). Opposite tendencies were found for the caproic acid (C6:0), and all analyzed factors and their interaction were statistically significant for this FA content in BCOL (source of colostrum *p* < 0.001, LAB strain used for fermentation *p* = 0.004, and interaction *p* = 0.013) (Appendix A). When comparing non-fermented and fermented sample groups, the highest content of C6:0 was found in the non-fermented D sample group (1.05% of the total fat content), in the E135 samples fermented with LUHS135 (1.05% of the total fat content), and in the D244 samples fermented with the LUHS244 groups (1.10% of the total fat content). A significant factor for the caprylic acid (C8:0) content in BCOL was the source of colostrum (*p* = 0.002) (Appendix A). When comparing the C8:0 content in non-fermented and fermented groups within the same source of colostrum, the highest content of C8:0 was found in the non-fermented A and D sample groups (0.182 and 0.340% of the total fat content, respectively), with B and C sample groups fermented with LUHS135 (0.303 and 0.309% of the total fat content, respectively), and in the E244 sample group fermented with LUHS244 (0.323% of the total fat content). The highest capric acid (C10:0) content was found in the A, D, C135, B244, and E244 samples (0.861, 1.19, 1.11, 1.04, and 1.08% of the total fat content, respectively), and analyzed factors (source of colostrum and LAB strain used for fermentation) and their interaction were statistically significant on C10:0 content in BCOL (*p* < 0.001) (Appendix A). Undecanoic (C11:0), tridecanoic (C13:0), *cis*-10-pentadecenoic (C15:1), and *cis*-10-heptadecenoic (C17:1) acids were detected only in the non-fermented sample groups and LAB strain used for fermentation was a statistically significant factor for this fatty acid content in BCOL (*p* < 0.001), as well as for C17:1 content in BCOL; moreover, all analyzed factors and their interaction were statistically significant (Appendix A). On the contrary, oleic (C18:1), linoleic (C18:2), α-linolenic (C18:3 α), and arachidonic (C20:4) acids were formed only in fermented sample groups, where their content was, on average, 17.1, 1.65, 0.399, and 0.234% of the total fat content, respectively. The LAB strain used for fermentation was shown to be a statistically significant factor in the fatty acid content in BCOL (*p* < 0.001) (Appendix A). Moreover, fermentation significantly increased palmitic (C16:0) and stearic (C18:0) acid contents in BCOL. When comparing the sample groups of non-fermented and fermented with LUHS135 and LUHS244, C16:0 content was, on average, 9.04, 38.1 and 38.0% of the total fat content, respectively, and C18:0 content was, on average, 0.342, 9.12 and 9.11% of the total fat content, respectively. Fermentation significantly reduced lauric acid (C12:0) content in BCOL: in sample groups A135, B135, C135, D135, and E135, such a reduction was, on average, 7.67, 5.85, 5.68, 5.70, and 5.65 times, respectively; additionally, in sample groups A244, B244, C244, D244, and E244 this reduction was, on average, 6.16, 5.92, 6.45, 5.66 and 5.56 times, respectively. The LAB strain used for fermentation was a statistically significant factor in C12:0 content in BCOL (*p* < 0.001) (Appendix A). Moreover, lower contents of myristoleic (C14:1), pentadecanoic (C15:0), palmitoleic (C16:1), and heptadecanoic (C17:0) acids were found in fermented sample groups when compared with non-fermented ones, and, in all these cases, the LAB strain used for fermentation was a statistically significant factor for the C14:1, C15:0, C16:1, and C17:0 contents in BCOL (*p* < 0.001) (Appendix A). Opposite tendencies for myristic acid (C14:0) were established and in all cases, significantly higher C14:0 content was found in the fermented sample groups, in comparison with non-fermented ones (in the sample groups A135, B135, C135, D135, and E135, on average, respectively, 23.8, 26.0, 20.9, 22.5 and 24.7 times higher; as well as in the sample groups A244, B244, C244, D244 and E244, on average, respectively, 23.8, 26.2, 20.1, 22.1, and 24.3 times higher). LAB strain used for fermentation proved to be a statistically significant factor regarding C14:0 content in BCOL (*p* < 0.001).

Despite the differences in individual fatty acid contents in BCOL, non-significant differences were found in the total saturated (SFA), monounsaturated (MUFA), and polyunsaturated (PUFA) fatty acids (Figure 20b). However, there were significant differences in the omega-3, 6, and 9 fatty acid contents (Figure 20c). The highest omega-3 FA content was found in the non-fermented B sample group (0.861% of the total fat content), and the highest omega-6 content was unveiled in the non-fermented A sample group (3.10% of the total fat content). Omega-9 fatty acids were predominant in BCOL, and their content ranged from, on average, 27.2% of the total fat content (in all non-fermented and fermented sample groups, except A135) to 28.1% of the total fat content (in the A135 group).

The fatty acid profile of BCOL differs from that of mature milk [195,196,197]. However, despite the existence of some reported studies, the data about FA profiles in BCOL are still scarce [197]. Sats et al. [197] reported that BCOL contains 64.9% of SFA, 31.4% of MUFA and 3.7% of PUFA. Moreover, they state that C16:0, C18:1 *cis*-9, and C18:2n-6 fatty acids are the main SFA, MUFA, and PUFA, respectively. These results are in line with the fatty acid profiles reported in earlier studies [195,196]. Our findings are in agreement with Contarini et al. [195] and O’Callaghan et al. [28], who revealed that the fatty acid constituents of BCOL were, on average, 70% saturated, 26% monounsaturated, and 4.5% polyunsaturated [28,195]. Despite the fact that PUFA are not the main fatty acid group in bovine colostrum, they were associated with anti-cancerogenic properties [198], and it was apparent that BCOL was a good source of these fatty acids [195]. Moreover, the predominant fatty acids in BCOL are palmitic (on average, 40% of the total fat content) and oleic (on average, 21% of the total fat content) acids [28]. However, a broad range of average values for fat content in BCOL can be seen [46,47]. Additionally, the profile of an individual fatty acid can vary in relation to the post-partum period and other factors. Laakso et al., Palmquist et al., and Lynch et al. reported that BCOL contains high concentrations of C18:0 and C18:1 fatty acids [196,199,200]. Many factors can be involved in the changes in fatty acid composition in BCOL, including the fermentation with LAB [14]. LAB fermentation may alter the fatty acid composition, increasing or decreasing certain fatty acids [201]. The variation in the fatty acid content of fermented bovine colostrum may occur due to excreted hydrolytic enzymes (lipases and esterases) by the LAB and the ability of these bacteria to utilize fatty acids for growth and synthesize of secondary metabolites [202,203]. It was also reported that LAB species use lipids in dairy products to generate short-chain, long-chain, and medium-chain fatty acids [204]. Some of the LAB strains can change the *cis*-9, *cis*-12 diene structure of C18 fatty acids (such as linoleic, α-linolenic and γ-linolenic acids) into the conjugated *cis*-9, *trans*-11, *trans*-9, and *trans*-11 diene structures [205]. However, the results of this study proved that the metabolic activity of LAB is also related to the composition of bovine colostrum samples (including chemical and microbial profiles). Finally, the control of the fermentation process is of foremost importance to ensure low variability on the fatty acid profile in BCOL. It is worth noting that when planning to process BCOL into high-value-added food, nutraceuticals, or pharmaceuticals, its composition needs to be adjusted during primary production. In this sense, animal-rearing technologies must be taken into consideration.

**Figure 20 animals-13-03154-f020:** (**a**–**c**) Fatty acid (FA) profile. (**a**)—individual FA; C4:0—butyric; C6:0—caproic; C8:0—caprylic; C10:0—capric; C11:0—undecanoic; C12:0—lauric; C13:0—tridecanoic; C14:0—myristic; C14:1—myristoleic; C15:0—pentadecanoic; C15:1—*cis*-10-pentadecenoic; C16:0—palmitic; C16:1—palmitoleic; C17:0—heptadecanoic; C17:1—*cis*-10-heptadecenoic; C18:0—stearic; C18:1—oleic; C18:2—linoleic; C18:3 α—α-linolenic; C20:4—arachidonic; (**b**)—SFA—saturated FA, MUFA—monounsaturated FA, PUFA—polyunsaturated FA; (**c**)—omega-3, 6, 9 of non-treated and fermented bovine colostrum (% from total fat content) [A—source of colostrum in Dauksiai, Lithuania, B—T source of colostrum in Sauselio village, Lithuania, C—source of colostrum in Geluvos village, Lithuania, D—source of colostrum in Paliepiu village, Lithuania, E—source of colostrum in Alksnupiu village, Lithuania; 135—fermented with *Lactiplantibacillus plantarum* strain 135; 244—fermented with *Lacticaseibacillus paracasei* strain 244; SFA saturated fatty acids, MUFA—monounsaturated fatty acids, PUFA—polyunsaturated fatty acids; Data were expressed as mean values ± SE. For each fatty acid group, different letters ^(a–d)^ indicate a significant difference (*p* ≤ 0.05) among all treatments.

## 4. Conclusions

Fermentation with the LAB strains *Lp. plantarum* 135 and *Lc. paracasei* 244 reduced the pH of bovine colostrum, on average, by 30.5%, respectively, in comparison with non-fermented samples. LAB viable counts were similar in non-fermented and fermented bovine colostrum. The results of essential and non-essential amino acids showed that the source of colostrum was a statistically significant factor in most of the detected amino acid content (except glutamine and proline). The LAB strain used for fermentation was a statistically significant factor in aspartic acid, threonine, glycine, alanine, methionine, phenylalanine, lysine, histidine, and tyrosine, and the factor interaction was statistically significant for the content of most detected amino acids (except glutamine, glutamic acid, valine, and leucine/isoleucine content).

Opposite trends were found for biogenic amine formation in bovine colostrum. Fermentation increased total biogenic amine content in the A135, A244, B135, and B244 sample groups (on average, by 85.7, 84.5, 48.4 and 47.0%, respectively); however, a decrease in the total BA content in the D135, D244, E135, and E244 samples was observed (on average, by 14.8, 27.7, 21.8 and 49.5%, respectively). Total biogenic amine content showed a significant correlation with glutamic acid, serine, aspartic acid, valine, methionine, phenylalanine, histidine, and GABA content. The source of colostrum was a statistically significant factor for total biogenic amine content in bovine colostrum.

Despite the differences in individual fatty acid contents in bovine colostrum, non-significant differences were found in the total saturated, monounsaturated, and polyunsaturated fatty acids. However, there were significant differences in omega-3, 6, and 9 fatty acid contents. The highest omega-3 fatty acid content was found in the non-fermented B samples group (0.861% of the total fat content), and the highest omega-6 content was disclosed in the non-fermented A samples group (3.10% of the total fat content). Omega-9 fatty acids were predominant in bovine colostrum, and their content ranged from, on average, 27.2% of the total fat content (in all non-fermented and fermented sample groups, except A135) to 28.1% of the total fat content (in the A135 group).

Overall, the utilization of bovine colostrum is challenging because of the sensitive compounds as well as the variability of their composition. Additionally, control of the fermentation process is of utmost importance to guarantee a low variability of the fermented bovine colostrum. In addition, the results here indicate that the source of bovine colostrum represents a major factor towards the success of the processing of colostrum to be applied in the food, nutraceutical, and pharmaceutical industries.

## Figures and Tables

**Figure 1 animals-13-03154-f001:** The experimental design.

**Table 1 animals-13-03154-t001:** The values of pH and lactic acid bacteria (LAB) viable counts (log_10_ CFU/mL) in bovine colostrum.

BSC	pH	LAB Viable Counts, log_10_ CFU/mL
Non-fermented bovine colostrum samples
A	6.41 ± 0.34 ^a,B^	7.89 ± 0.35 ^a,A^
B	6.22 ± 0.39 ^a,B^	8.01 ± 0.36 ^a,A^
C	6.32 ± 0.42 ^a,B^	8.04 ± 0.80 ^a,A^
D	6.32 ± 0.35 ^a,B^	7.92 ± 0.66 ^a,A^
E	6.25 ± 0.32 ^a,B^	7.92 ± 0.33 ^a,A^
Overall mean	6.30 ± 0.03	7.96 ± 0.03
Fermented with LAB strain 135 bovine colostrum samples
A_135_	4.67 ± 0.28 ^a,A^	8.37 ± 0.57 ^a,A^
B_135_	4.52 ± 0.35 ^a,A^	8.33 ± 0.27 ^a,A^
C_135_	4.22 ± 0.31 ^a,A^	8.37 ± 0.28 ^a,A^
D_135_	4.40 ± 0.37 ^a,A^	8.37 ± 0.79 ^a,A^
E_135_	4.18 ± 0.32 ^a,A^	8.38 ± 0.39 ^a,A^
Overall mean	4.40 ± 0.09	8.36 ± 0.01
Fermented with LAB strain 244 bovine colostrum samples
A_244_	4.61 ± 0.27 ^a,A^	8.31 ± 0.26 ^a,A^
B_244_	4.58 ± 0.44 ^a,A^	8.29 ± 0.82 ^a,A^
C_244_	4.35 ± 0.39 ^a,A^	8.34 ± 0.57 ^a,A^
D_244_	4.21 ± 0.30 ^a,A^	8.33 ± 0.59 ^a,A^
E_244_	4.10 ± 0.31 ^a,A^	8.26 ± 0.43 ^a,A^
Overall mean	4.37 ± 0.10	8.31 ± 0.01

BSC—bovine colostrum samples; A—source of colostrum in Dauksiai village, Lithuania. B—T source of colostrum in Sauselio village, Lithuania. C—source of colostrum in Geluvos village, Lithuania. D—source of colostrum in Paliepiu village, Lithuania. E—source of colostrum in Alksnupiu village, Lithuania; 135—*Lactiplantibacillus plantarum* strain 135; 244—*Lacticaseibacillus paracasei* strain 244. The data were expressed as mean values ± SE. ^a^—means within a column with different superscript letters are significantly different (*p* ≤ 0.05) for the same bovine colostrum sample group (non-fermented, fermented with LAB strain 135, and fermented with LAB strain 244); ^A,B^—means within a column with different superscript letters are significantly different (*p* ≤ 0.05) for the same farm (A, B, C, D, and E).

## Data Availability

Not applicable.

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
