# Peer review of "Ascertaining the Influence of Lacto-Fermentation on Changes in Bovine Colostrum Amino and Fatty Acid Profiles"

_animals, 2023, doi:10.3390/ani13193154_

Round 1

Reviewer 1 Report

Please check all note and comments and revise accordingly.

The manuscript is potentially worth to be published as it illustrated a new and relatively valuable information on bovine colostrum.

However, the manuscript requires major revision as highlighted throughout the manuscript and summarized the main comments below:

Avoid long sentences. Ex lines 114-117,

Better to use passive voice in scientific writing

Avoid repeating same information. Line 107-109, 139-140

Please check and revise experimental design. There is some contradiction in you description. Section 2.2 and line 228.

Be careful with the statement descripting significant and insignificant differences. I have highlighted many. Ex. Line 244, 251.

The statements describing the means at the bottom of tables are not clear and must be revised. Ex Table 1.

All Figs should be labelled more properly, columns with each treatment should be in different colour or pattern.

Avoid repeating SE- standard error. In each table. Once is enough.

Spell out the genus name only the first time, then The 1st capital letter only.

Check the tenses throughout the manuscript.

The manuscript included too many tables (6) and Figs (20). May be better to list some of these as appendixes.

Check references and apply same format to all. (Year in bold). Ex ref 122,79, 52.

Needs some improvement, especially, when reporting significant and insignificant difference.

Author Response

Dear Reviewer,

We would like to thank you and the reviewers for your time spent on reviewing our manuscript entitled “Ascertaining the influence of lacto-fermentation on changes in bovine colostrum amino and fatty acid profiles”, and your insightful comments helping us improving the work.

Authors

Reviewer 2 Report

General comments:

Less is more.

The manuscript dealt with the potential effect of lacto-fermentation on bovine colostrum components. The authors have taken on the enormous task to process such vast amounts of data and interpret the results, that showed no clear tendency. It seems to me that there were no a priori research hypothesis as to what is expected to happen to the e.g. amino acid and fatty acid profiles and authors somewhat struggled to interpret the effects of lacto-fermentation that seemed to be quite random. The manuscript is extremely long and very tiring to read. Reducing to some major points of interest is strongly advised.

It is also not clear how the correlation between samples obtained from the same farm and the even more obvious correlation of results deriving from the same colostrum sample were taken into consideration during analysis. If it was not taken into consideration, the results are not well-founded and analysis with a proper method could, in fact, hugely change the p-values.

Detailed comments:

Introduction: very good, some minor comments:

Line 57: Is this about colostrum in general, or bovine colostrum? Please specifiy, if the latter holds.

Line 90 and elsewhere: authors like to use the term 'representative'. It needs to be please specified in what aspect is sample collection considered representative Geographically, according to breeds, according to age/parity of the cow population?

Line 107: This is meant to be a final, or second aim of the study, however it sounds exactly the same as main aim described in Line 90-92. Could you please specificy the difference between primary and secondary objectives.

Materials and methods: generally well written, needs some supplementation.

First: Why it is so that other bacterial strains were not sought for? In the untreated BCS samples, there was no drop of pH to prevent other microbes to grow.

Line 125: Could you please use another abbreviation for bovine colostrum? BCS is the widely accepted abbreviation of body condition score and it can be very distracting. I suggest using BCOL or simply BC as in your previous articles on the subject. Thank you.

Line 151-155: There is absolutely no information on dairy management technologies of the farms, however, authors base the conclusions on presumed, but seemingly unrecorded, differences.

Indeed, colostrum quality can depend on many factors. Thus, it is crucial to know how the 5% of cows were selected. Did the sampled group include both primiparous and multiparous cows? Was it collected in different seasons? Were the breeds different? Please describe in a few words the strategy for choosing the cows.

Statistical analysis: As I have already mentioned in the introductory words, it is not clear to me how correlatedness of samples from the same farm, and results from the same sample treated three ways were managed. It is also stated in Line 229-230 that analyses were done in triplicate. I am pretty sure you are not talking about statistical analysis done in triplicate :) If it is three paralell measurements of the same sample, this has to be written in previous parts of the mat and meth section.

I would suggest using generalised least squares or even better, mixed effects model (there are alternative names, like random effects model, multilevel model etc.) This way the effect of farms would not tie down valuable degrees of freedom! Farm should be included as a random factor. Now farm is included as explanatory variable though the effects of farms were not compared, interpreted or discussed in any way. Also, at the current state the results are only applicable to these 5 farms, but I suppose that you wished your results to be generalizable to dairy cows in general. Could you please explain why you wished to quantify the effect of 'farm'?

Why I am especially keen on the correlation of samples is that many many data from repeated measures of the same few samples does not hold as much information as many many data from independent samples. Repeated measurements (even it means different treatments of the same colostrum sample) entail a lower effective sample size. This has to be taken into consdieration when calculating standard errors. A lower effective sample size increases standard error. Higher standard errors lead to higher p-values. If correlatedness of measurements are not taken into account, the chance of commiting type I error increases manyfold. Unless you provide the necessary explanations on how correlatedness was included in the model, I can not accept the results as reliable. If it was not included then tests have to be rerun with an appropriate analysis. Using MANOVA does not address this issue.

I suppose once the proper analysis is done, at least half of the significant differences will vanish and that reduces the amount of results to a readable size.

Pearson correlations require independent observations and a linear relationship. Are you positive that these hold? May I refer to the excellent works of Bland and Altman on repeated measures correlation?

Results

Line 237: Results should be renamed to Results and discussion.

Line 242-244: includes an unnecessary repetition of words.

Line 245-249. Hard to comprehend. Please transfer to Table 1 as overall means in new rows.

Line 258-285. These statements are a bit off topic. It does not answer why the pH of uninoculated samples were higher than inoculated samples, however having a similar LAB CFU count. Please present relevant statements.

Line 267-268: If substrate-specificity issuch a big issue, why was colostrum components not analyzied in the first place to exclude outliers (potential confounding)? Anyway, the farms did not differ from each other, which means there were no great differences in lactose contents.

L274-276: About the cell counts, it seems that LAB counts reached the stationery phase in all cultures. Indeed, the cell count approaches that of the strain cultures (9.2 log10 CFU/ml).  It is not that surprising that no differences were measured.

Table 1. 135 missing after 'Fermented with LAB strain' in middle line.

Table 1. caption : "within a line": please specifiy to "within a row" or "within a column", respectively.

Line 287. The 3.2. subchapter should be separated regarding each of the amino acids (e.g. 3.2.1. Phe, 3.2.2. Val and so on).

Line 354, 355 and elsewhere: Once the interaction of two factors is significants, the single effect of any of the factors can not be in itself interpreted, the interaction determines the effects regarding the level of the factors considered. It is almost impossible to explain the interaction of a farm and a bacterial strain. It once again urges a baseline comparison of farms, regarding colostrum constituents. I am afraid you have no leftover samples, though. If so, I strongly advise you to do so.

In general, I find all parts discussing the changes of the amino acid concentration to be more like parts of a university biology book instead of plausible explanations of the observed differences. I am surprised that after the excessive studying of the used LUHS135 and LUH244 strains (as the cited publications illustrate it), the authors did not have an a priori idea of differences in bacterial enzyme activities and fermentation capabilities.

Table 2. See earlier comments on interaction. Pariwise comparisons are needed to detect differences, but it becomes such a mess with so many levels, and any interpretation is hard to be given.

Table 4. Consider leaving out entirely. In my opinion, the significant p-values occurred purely by chance, conducting significance hunting at such an enormous scale. Did you consider adjusting for multiple comparisons in the first place??

Line 808-822. Seems a bit irrelevant to me. Did you consider the proteolytic effect of LAB or other bacteria present?

What was your hypothesis on the effect of LAB on BAs? Sorry if I missed something during reading.

It seems that LAB inoculation increased total BA content. Would you still recommend using these strains as preprocessing?

Figure 20 a) It is striking how big a difference there is between untreated and treated samples. Unfortunately the legend colors are impossible to read. Could you please improve it a bit?

Where is the reason explained for such huge changes in c12:0, c14:1 and c18:2 concentrations? Once again, sorry if I missed it.

From Line 955: The conclusions are more like results, boiling down to the effect of animal rearing technologies that were not even investigated in the first place. Consider revising.

Wrong word order, repetitions, stylistic mistakes here and there. I consider using a language check software with a general tone isntead of academic.

Author Response

Dear Reviewer:
We would like to thank you and the reviewers for your time spent on reviewing our manuscript entitled “Ascertaining the influence of lacto-fermentation on changes in bovine colostrum amino and fatty acid profiles”, and your insightful comments helping us improving the work.

The Authors

Round 2

Reviewer 1 Report

Significant improvement. However, I have one more comment regarding the Figs. foot notes. The statement: Different letters (a–f) indicate a significant difference (p ≤ 0.05), is not descriptive enough. are you refereeing to all treatments, or within each treatment?

Please make it clear.

Different letters (a–f) indicate significant differences (p ≤ 0.05) among all treatments.

or

Different letters (a–f) indicate significant differences (p ≤ 0.05) between groups within each treatment.

Author Response

(The authors gave the same response as above.)

Reviewer 2 Report

Dear Authors,

Thank you for taking the time to respond to an excrutiangly long row of comments.

I found it easiest to re-respond to all responses in the previously sent document containing the first report. Please find it attached. New inclusions have green as font colour.

Author Response

(The authors gave the same response as above.)
